# Adaptive Significance Levels in Tests for Linear Regression Models: The *e*-Value and *P*-Value Cases

**DOI:** 10.3390/e25010019

**Published:** 2022-12-22

**Authors:** Alejandra E. Patiño Hoyos, Victor Fossaluza, Luís Gustavo Esteves, Carlos Alberto de Bragança Pereira

**Affiliations:** 1Facultad de Ingeniería, Institución Universitaria Pascual Bravo, Medellín 050034, Colombia; 2Instituto de Matemática e Estatística, Universidade de São Paulo, São Paulo 05508-090, Brazil

**Keywords:** adaptive significance levels, Bayesian test, linear regression, predictive distribution, significance test

## Abstract

The full Bayesian significance test (FBST) for precise hypotheses is a Bayesian alternative to the traditional significance tests based on *p*-values. The FBST is characterized by the *e*-value as an evidence index in favor of the null hypothesis (**H**). An important practical issue for the implementation of the FBST is to establish how small the evidence against **H** must be in order to decide for its rejection. In this work, we present a method to find a cutoff value for the *e*-value in the FBST by minimizing the linear combination of the averaged type-I and type-II error probabilities for a given sample size and also for a given dimensionality of the parameter space. Furthermore, we compare our methodology with the results obtained from the test with adaptive significance level, which presents the capital-P *P*-value as a decision-making evidence measure. For this purpose, the scenario of linear regression models with unknown variance under the Bayesian approach is considered.

## 1. Introduction

The full Bayesian significance test (FBST) for precise hypotheses is presented in [1] as a Bayesian alternative to the traditional significance tests based on *p*-values. With the FBST, the authors introduce the *e*-value as an evidence index in favor of the null hypothesis (**H**). An important practical issue for the implementation of the FBST is to establish how small the evidence must be to decide to reject **H** ([2,3]). In that sense, the authors of [4] present loss functions such that the minimization of their posterior expected values characterizes the FBST as a Bayes test under a decision-theoretic approach. This procedure provides a cutoff point for the evidence that depends on the severity of the error for deciding whether to reject or accept **H**.

In the frequentist significance-test context, it is known that under certain conditions the *p*-value decreases as the sample size increases, in such a way that by setting a single significance level, the comparison of the *p*-value with the fixed significance level usually leads to rejection of the null hypothesis ([5,6,7,8,9]). In the FBST procedure, the *e*-value exhibits similar behavior to the *p*-value when the sample size increases, which suggests that the cutoff point to define the rejection of **H** should depend on the sample size and (possibly) on other characteristics of the statistical model under consideration. However, in the proposal of [4], a loss function that explicitly takes into account the sample size is not studied.

In order to solve the problem of testing hypotheses in the usual way, in which changing the sample size influences the probability of rejecting or accepting the null hypothesis, the authors of [10], motivated by [11], suggest that the level of significance in hypothesis testing should be a function of sample size. Instead of setting a single level of significance, the authors of [10] propose fixing the ratio of severity between type-I and type-II error probabilities based on the incurred losses in each case, and thus, given a sample size, defining the level of significance that minimizes the linear combination of the decision error probabilities. The authors of [10] show that proceeding this way, by increasing the sample size, the probabilities of both kind of errors and their linear combination decrease, while in most cases, setting a single level of significance independent of sample size, only type-II error probability decreases. The tests proposed in [10] take the same conceptual grounds of the usual tests for simple hypotheses based on the minimization of a linear combination of probabilities of error of decisions as presented in [12]. Then, the authors of [10] extend, in a sense, the idea in [12] to composite and sharp hypotheses, according to the initial work in [11].

Following the same line of work, the authors of [13,14] present a new hypothesis-testing procedure formulated from the ideas developed in previous works ([11,15,16,17]) and using a mixture of frequentist and Bayesian tools. This procedure introduces the capital-P *P*-value as a decision-making evidence measure and also includes an adaptive significance level, i.e., a significance level that is a function of sample size. Such an adaptive significance level is obtained from the minimization of the linear combination of generalized type-I and type-II error probabilities. According to the authors of [14], the resulting hypothesis tests do not violate the likelihood principle and do not require any constraints on the dimensionalities of the sample space and parameter space. It should be noticed that the new test procedure is precisely the optimal decision rule for the problem of testing the simple hypotheses $f_{H}$ against $f_{A}$. For this reason, such a procedure overcomes the drawback of increasing the sample size resulting in the rejection of a null precise hypothesis ([12]). Another important way of successfully dealing with this question is to take into account meaningful deviations from the parameter value that specifies the null precise hypothesis in the formulation of the hypothesis testing problem ([18,19]).

On the other hand, linear models are probably the most used statistical models to establish the influence of a set of covariates on a response variable. In that sense, the proper identification of the relevant variables in the model is an important issue in any scientific investigation and is a more challenging task in the context of Big-Data problems. In addition to high dimensionality, in recent statistical learning problems, it is common to find large datasets with thousands of observations. This fact may cause the hypothesis of nullity of the regression coefficients to be rejected most of the time, due to the large sample size when the significance level is fixed.

The main goal of our work is to determine, in the setting of linear regression models, how small the Bayesian evidence in the FBST should be in order to reject the null hypothesis and prevent a decision-maker from the abovementioned drawbacks. Therefore, taking into account the concepts in [11,12] associated with optimal hypothesis tests, as well as the conclusions in [10] about the relationship between the significance levels and the sample size, and finally, considering the ideas developed recently by the authors of [13,14] related to adaptive significance levels, we present a method to find a cutoff point for the *e*-value by minimizing a linear combination of the averaged type-I and type-II error probabilities for a given sample size and also for a given dimensionality of the parameter space. For that purpose, the scenario of linear regression models with unknown variance under the Bayesian approach is considered. So, by providing an adaptive level for decision making and controlling the probabilities of both kinds of errors, we intend to avoid the problems associated with the rejection of the hypotheses on the regression coefficients when the sample size is very large. In addition to the *e*-value, we calculate the *P*-value as well as its corresponding adaptive significance levels in order to compare the decisions that can be made by performing the tests with each of these measures.

## 2. The Linear Regression Model with Unknown Variance

The identification of the relevant variables in linear models can be done through hypothesis-testing procedures involving the respective regression coefficients. In the conjugate Bayesian analysis of the normal linear regression model with unknown variance, it is possible to obtain expressions for the posterior distributions of the parameters and their respective marginals. Therefore, in this setting, the FBST can be used for testing if one or more of the regression coefficients is null, which is the basis of one possible model-selection procedure. We first review the normal linear regression model
(1)$$y=X\theta +\varepsilon ,\;\varepsilon ∼N_{n}\left(0,\sigma^{2}I_{n}\right),$$
where $y={\left(y_{1},\cdots ,y_{n}\right)}^{⊤}$ is an n×1 vector of $y_{i}$ observations, $X={\left(x_{1},\cdots ,x_{n}\right)}^{⊤}$ is an n×p matrix of covariates, also called the design matrix, with $x_{i}={\left(1,x_{i1},\cdots ,x_{ip-1}\right)}^{⊤}$, $\theta ={\left(\theta_{1},\cdots ,\theta_{p}\right)}^{⊤}$ is a p×1 vector of parameters (regression coefficients), and $\varepsilon ={\left(\varepsilon_{1},\cdots ,\varepsilon_{n}\right)}^{⊤}$ an n×1 vector of random errors. The model shows simply that the conditional distribution of y given parameters $\left(\theta ,\sigma^{2}\right)$ is the multivariate normal distribution $N_{n}\left(X\theta ,\sigma^{2}I_{n}\right)$. Therefore, the likelihood becomes
(2)$$f\left(y|\theta ,\sigma^{2}\right)={\left(2\pi \sigma^{2}\right)}^{-n/2}\exp\left\{-\frac{1}{2\sigma^{2}}{\left(y-X\theta \right)}^{⊤}\left(y-X\theta \right)\right\}.$$

The natural conjugate prior distribution of $\left(\theta ,\sigma^{2}\right)$ is a *p*-variate normal-inverse gamma distribution with hyperparameters $m_{0}$, $V_{0}$, $a_{0}$, and $b_{0}$, denoted by $\left(\theta ,\sigma^{2}\right)∼N_{p}IG\left(m_{0},V_{0},a_{0},b_{0}\right)$. Combining it with the likelihood (Equation 2) gives the posterior distribution ([20,21,22]):(3)$$f\left(\theta ,\sigma^{2}|y\right)\propto {\left(\sigma^{2}\right)}^{-\left(a_{0}+\frac{n}{2}+\frac{p}{2}+1\right)}\;\begin{array}{c} \exp\left\{-\frac{1}{2\sigma^{2}}\left[{\left(\theta -m^{*}\right)}^{⊤}\;V^{*}-1\;\left(\theta -m^{*}\right)+2b_{1}\right]\right\} \end{array},$$
where
$$V^{*}={\left({V_{0}}^{-1}+X^{⊤}X\right)}^{-1},\;m^{*}=V^{*}\;\left({V_{0}}^{-1}m_{0}+X^{⊤}y\right),$$
a1=a0+n2,b1=b0+m0V0−1⊤m0+y⊤y−m*V*−1⊤m*2.

If $X^{⊤}X$ is non-singular, we can write
$$m^{*}=V^{*}\;\left({V_{0}}^{-1}m_{0}+X^{⊤}X\hat{\theta }\right),$$
where $\hat{\theta }={\left(X^{⊤}X\right)}^{-1}X^{⊤}y$ is the classical maximum likelihood or least squares estimator of θ. Therefore, the posterior distribution of $\left(\theta ,\sigma^{2}\right)$ is
$$\left(\theta ,\sigma^{2}\right)|y∼N_{p}IG\left(m^{*},V^{*},a_{1},b_{1}\right).$$

See Appendix A for further explanation of the priors, posteriors, and conditional distributions for the linear regression models with unknown variance.

## 3. Adaptive Significance Levels in Linear Regression Coefficient Hypothesis Testing

In this section, we present the methodology to find a cutoff value for the evidence in the FBST as an adaptive significance level and we also develop the procedure to calculate the *P*-value with its corresponding adaptive significance level, all this in the context of linear regression coefficient hypothesis testing in models with unknown variance under the Bayesian point of view. For that purpose, first of all, it is necessary to show how the Bayesian prior predictive densities under the null and alternative hypotheses are defined.

### 3.1. Prior Predictive Densities in Regression-Coefficient Hypothesis Testing

Let $\theta ={\left(\theta_{1}^{⊤}\theta_{2}^{⊤}\right)}^{⊤}$, with $\theta_{1}={\left(\theta_{1},\cdots ,\theta_{s}\right)}^{⊤}$ and $\theta_{2}={\left(\theta_{s+1},\cdots ,\theta_{p}\right)}^{⊤}$, having $\theta_{1}$*s* elements and $\theta_{2}$*r* elements. Let $\xi ={\left(\theta^{⊤},\sigma^{2}\right)}^{⊤}={\left(\theta_{1}^{⊤},\theta_{2}^{⊤},\sigma^{2}\right)}^{⊤}$, then, $Y|\xi ∼N_{n}\left(X\theta ,\sigma^{2}I_{n}\right)$ where ξ∈Ξ. We are interested in testing the hypotheses
$$\begin{array}{c} H:\theta_{2}=0\\ A:\theta_{2}\neq 0. \end{array}$$

Let $\Xi_{H}$ and $\Xi_{A}$ be the partition of the parameter space defined by the competing hypotheses **H** and **A**. Consider the prior density g(ξ) defined over the entire parameter space Ξ and let $f_{H}$ and $f_{A}$ be the Bayesian prior predictive densities under the respective hypotheses. Both are probability density functions over the sample space Ω, as follows:(4)$$\begin{array}{cc} f_{H}\left(y\right)= & \;\;t_{n}\left(2\left(a_{0}+\frac{r}{2}\right);XCm_{0_{1.2}}\left(0\right),\frac{b_{0}+\frac{{m_{0_{2}}}^{⊤}\;{\left(V_{0_{22}}\right)}^{-1}m_{0_{2}}}{2}}{\left(a_{0}+\frac{r}{2}\right)}\left(I_{n}+\left(XC\right)V_{0_{11.2}}{\left(XC\right)}^{⊤}\right)\right), \end{array}$$
where $C_{(s+r)\times s}={\left[I_{s},0_{s\times r}\right]}^{⊤}$.

Additionally,
(5)$$\begin{array}{cc} f_{A}\left(y\right)\; & =t_{n}\left(2a_{0};Xm_{0},\frac{b_{0}}{a_{0}}\left(I_{n}+XV_{0}X^{⊤}\right)\right). \end{array}$$
where $P_{H}$ and $P_{A}$ are the prior probability measure of ξ restricted to the sets H and A respectively (more details can be seen in Appendix B).

### 3.2. Evidence Index: e-Value

The *full Bayesian significance test* (FBST) was proposed in [1] for precise or “sharp” hypotheses (subsets of the parameter space with smaller dimension than the dimension of the whole parameter space, and, therefore, with null Lebesgue measure) based on the evidence in favor of the null hypothesis, calculated as the posterior probability of the complement of the highest posterior density (HPD) region (here we consider the usual HPD region with respect to the Lebesgue measure, even though it could be built by choosing any other dominating measure instead) tangent to the set that defines the null hypothesis. Considering the concepts in [10,11], and the recent works [13,14] related to adaptive significance levels, we propose to establish a cutoff value $k^{*}$ for the e-value ($ev\left(H;y_{0}\right)$) in the FBST as a function of the sample size *n* and the dimensionality of the parameter space *d*, i.e., $k^{*}=k^{*}\left(n,d\right)$ with $k^{*}\in \left[0,1\right]$, such that $k^{*}$ minimizes the linear combination of the averaged type-I and type-II error probabilities, aα+bβ. To describe the procedure in the context of the coefficient hypothesis testing of the linear regression model we are addressing, consider the tangential set to the null hypothesis which is defined as
(6)$$\begin{array}{cc} T_{y_{0}} & =\left\{\xi \in \Xi :f\left(\xi |y_{0}\right)>\underset{H}{\sup}f\left(\xi |y_{0}\right)\right\}\\ \; & =\left\{\left(\theta_{1},\theta_{2},\sigma^{2}\right)\in \Xi :f\left(\theta_{1},\theta_{2},\sigma^{2}|y_{0}\right)>\underset{H}{\sup}f\left(\theta_{1},\theta_{2},\sigma^{2}|y_{0}\right)\right\}. \end{array}$$

This is the posterior distribution of $\left(\theta_{1},\sigma^{2}\right)$ given $\theta_{2}$ a *s*-variate normal-inverse gamma, that is
$$\left(\theta_{1},\sigma^{2}|\theta_{2},y_{0}\right)∼N_{s}IG\left({m^{*}}_{1.2}\left(\theta_{2}\right),{V^{*}}_{11.2},a_{0}+\frac{r}{2},b_{0}+\frac{{\left(\theta_{2}-{m^{*}}_{2}\right)}^{⊤}\;{V^{*}}_{22}^{-1}\left(\theta_{2}-{m^{*}}_{2}\right)}{2}\right),$$
where the point under **H** for which the posterior attains its maximum value can be calculated as follows
$$\begin{array}{cc} \underset{H}{arg sup}f\left(\theta_{1},\theta_{2},\sigma^{2}|y_{0}\right) & =\underset{\theta_{1},\theta_{2}=0,\sigma^{2}}{arg sup}f\left(\theta_{1},\theta_{2}=0,\sigma^{2}|y_{0}\right)\\ \; & =\underset{\theta_{1},\sigma^{2}}{arg sup}\;\;\frac{f(\theta_{1},\theta_{2}=0,\sigma^{2}|y_{0})}{\int_{\theta_{1}\in R^{s},\;\sigma^{2}\in R_{+}}f\left(\theta_{1},\theta_{2}=0,\sigma^{2}|y_{0}\right)\;\;d\theta_{1}\;d\sigma^{2}}\\ \; & =\underset{\theta_{1},\sigma^{2}}{arg sup}f\left(\theta_{1},\sigma^{2}|\theta_{2}=0,y_{0}\right)\\ \; & =\mathrm{Mode}\left[f(\theta_{1},\sigma^{2}|\theta_{2}=0,y_{0})\right]\\ \; & =\left[{m^{*}}_{1.2}\left(\theta_{2}=0\right),\;0,\;\frac{b_{1}+\frac{{\left({m^{*}}_{2}\right)}^{⊤}\;{\left({V^{*}}_{2}\right)}^{-1}\left({m^{*}}_{2}\right)}{2}}{\left(a_{1}+\frac{r}{2}\right)+1+\frac{s}{2}}\right]\\ \; & =\left[\hat{\theta_{1}},\;0,\;\hat{\sigma^{2}}\right]. \end{array}$$

Thus, we get the tangential set
(7)$$\begin{array}{cc} T_{y_{0}}\; & =\left\{\left(\theta_{1},\theta_{2},\sigma^{2}\right)\in \Xi :f\left(\theta_{1},\theta_{2},\sigma^{2}|y_{0}\right)>f\left(\hat{\theta_{1}},0,\hat{\sigma^{2}}|y_{0}\right)\right\}. \end{array}$$

The evidence in favor H is calculated as the posterior probability of the complement of $T_{y_{0}}$. That is,
(8)$$ev\left(H;y_{0}\right)=1-P\left(\xi \in T_{y_{0}}|y_{0}\right).$$

The evidence index, *e*-value, in favor of a precise hypothesis, considers all points of the parameter space which are less “probable" than some point in $\Xi_{H}$. A large value of $ev\left(H;y_{0}\right)$ means that the subset $\Xi_{H}$ lies in a high-probability region of Ξ, and, therefore, the data support the null hypothesis; on the other hand, a small value of $ev\left(H;y_{0}\right)$ means that $\Xi_{H}$ is in a low-probability region of Ξ and the data would make us discredit the null hypothesis ([23]).

The evidence in (Equation 8) can be approximately determined via Monte Carlo simulation. Then, generating *M* samples from the posterior distribution of ξ, such that $\xi |y∼N_{p}IG\left(m^{*},V^{*},a_{1},b_{1}\right)$, we estimate the evidence by Monte Carlo simulation through the expression
$$1-\frac{1}{M}\sum_{j=1}^{M}1\left(\xi^{\left(j\right)}\in T_{y_{0}}\right).$$

Now, consider the test such that
$$\varphi_{e}\left(y\right)=\left\{\begin{array}{c} 0\;if\;ev\left(H;y\right)>k\\ 1\;if\;ev\left(H;y\right)\leq k. \end{array}\right.\;$$

The averaged error probabilities, expressed in terms of the predictive densities, can be estimated by Monte Carlo simulation through the expressions
(9)$$\alpha_{\varphi_{e}}=\int_{y\in \Psi_{e}}f_{H}\left(y\right)\;dy\;\mathrm{and}\;\beta_{\varphi_{e}}=\int_{y\notin \Psi_{e}}f_{A}\left(y\right)\;dy,$$
where $\Psi_{e}$ is the set
$$\Psi_{e}=\left\{y\in \Omega :ev\left(H;y\right)\leq k\right\}.$$

So, the adaptive cutoff value $k^{*}$ for $ev\left(H;y\right)$ will be the *k* that minimizes $a\alpha_{\varphi_{e}}+b\beta_{\varphi_{e}}$. The *a* and *b* values represent the relative seriousness of errors of the two types or, equivalently, relative prior preferences for the competing hypotheses. For example, if b/a=1, it is said that $\beta_{\varphi_{e}}$ and $\alpha_{\varphi_{e}}$ are equally severe, whereas if b/a<1, then $\alpha_{\varphi_{e}}$ undergoes a more intense minimization than $\beta_{\varphi_{e}}$, which means that type-I error is considered more serious than type-II error and also indicates a prior preference for **H**.

### 3.3. Significance Index: P-Value

The authors of [13,14] present a new hypothesis-testing procedure using a mixture of frequentist and Bayesian tools. On the one hand, the procedure resembles a frequentist test as it is based on the comparison of the *P*-value as a decision-making evidence measure with an adaptive significance level. On the other hand, such an adaptive significance level is obtained from the minimization of a linear combination of generalized type-I and type-II error probabilities under a Bayesian perspective. As a result, it generally depends on both the null and alternative hypotheses and on the sample size *n*, as opposed to standard fixed significance levels. The new proposal may also be seen as a test for simple hypotheses characterized by the predictive distributions $f_{H}$ and $f_{A}$ in Section 3.1 that minimizes a specific linear combination of probabilities of errors of decision. It is then formally characterized by a cutoff for the Bayes Factor (which takes the place of the likelihood ratio here) and therefore may prevent a decision-maker from rejecting the null hypothesis when the data seem to be clear evidence in its favor ([12]). It should be stressed that under the new proposal, a cutoff value for the Bayes factor (for the “likelihood ratio” here) is chosen in advance and consequently no constraint is imposed exclusively on the probability of the error of the first kind. In this sense, the test in [13,14] completely departs from regular frequentist tests. From another angle, the Bayes factor may be seen as the ratio between the posterior odds in favor of the null hypothesis and its prior odds ([24]). Note that the quantity defined here is a capital-P “*P*-value” to distinguish it from the small-p “*p*-value”. In the scenario of the linear regression model with unknown variance, the ratio between the two prior predictive densities (Equation 4) and (Equation 5), will be the Bayes factor,
(10)$$\mathrm{BF}\left(x\right)=\frac{f_{H}\left(x\right)}{f_{A}\left(x\right)}.$$

Now, consider the test
$$\varphi^{*}\left(y\right)=\left\{\begin{array}{c} 0\;if\;\mathrm{BF}\left(y\right)>\frac{b}{a}\\ 1\;if\;\mathrm{BF}\left(y\right)\leq \frac{b}{a}. \end{array}\right.\;$$

For any other test φ, $\varphi^{*}$ minimizes a linear combination of the type-I and type-II error probabilities, $a\alpha_{\varphi }+b\beta_{\varphi }$. Here again, the *a* and *b* values represent the relative seriousness of errors of the two types. To obtain the *P*-value at the point $y_{0}\in \Omega$, define the set $\Psi_{0}$ of sample points y for which the Bayes factors are smaller than or equal to the Bayes factor of the observed sample point $y_{0}$, that is
$$\Psi_{0}=\left\{y\in \Omega :\mathrm{BF}\left(y\right)\leq \mathrm{BF}\left(y_{0}\right)\right\}.$$

Then, the *P*-value is the integral of the predictive density over **H**, $f_{H}$, in $\Psi_{0}$
$$\begin{array}{c} P-\mathrm{value}\;\left(y_{0}\right)=\int_{\Psi_{0}}f_{H}\left(y\right)\;dy. \end{array}$$

Defining the set $\Psi^{*}$ of sample points y with Bayes factors smaller than or equal to b/a, i.e.,
$$\Psi^{*}=\left\{y\in \Omega :\mathrm{BF}\left(y\right)\leq \frac{b}{a}\right\},$$
the optimal averaged error probabilities from the generalized Neyman–Pearson Lemma, which will depend on the sample size, are given by
$$\alpha_{\varphi^{*}}=\int_{y\in \Psi^{*}}f_{H}\left(y\right)\;dy\;\mathrm{and}\;\beta_{\varphi^{*}}=\int_{y\notin \Psi^{*}}f_{A}\left(y\right)\;dy.$$

In order to make a decision, the *P*-value is compared to the optimal adaptive significance level $\alpha_{\varphi^{*}}$. Then, when $y_{0}$ is observed, the hypothesis **H** will be rejected if the $P-\mathrm{value}\;\left(y_{0}\right)<\alpha_{\varphi^{*}}$.

## 4. Simulation Study

We developed a simulation study considering two models. The first model was
(11)$$y=X\theta +\varepsilon ,\;\varepsilon ∼N_{n}\left(0,\sigma^{2}I_{n}\right),$$
where $X=1_{n}$ and $\theta =\theta_{1}$. The hypotheses to be tested were
$$\begin{array}{c} H:\theta_{1}=0\\ A:\theta_{1}\neq 0. \end{array}$$

The second model studied was
(12)$$y=X\theta +\varepsilon ,\;\varepsilon ∼N_{n}\left(0,\sigma^{2}I_{n}\right),$$
where $X={\left(x_{1},\cdots ,x_{n}\right)}^{⊤}$ is an n×p matrix of covariates with $x_{i}={\left(1,x_{i1},\cdots ,x_{ip-1}\right)}^{⊤}$ and $\theta ={\left(\theta_{1}^{⊤},\theta_{2}^{⊤}\right)}^{⊤}$ is the p×1 vector of coefficients. In this case, the hypotheses of interest were
$$\begin{array}{c} H:\theta_{2}=0\\ A:\theta_{2}\neq 0. \end{array}$$

The averaged error probabilities, $\alpha_{\varphi^{*}}$ and $\beta_{\varphi^{*}}$, were calculated using the Monte Carlo method with values generated from the following distributions:**Model (Equation 11) under H**$$\begin{array}{cc} \theta_{1}^{\left(j\right)} & =0\\ \sigma^{2(j)} & |\theta_{1}^{\left(j\right)}=0∼IG\left(a_{0}+\frac{1}{2},b_{0}+\frac{{\left(\theta_{1}^{\left(j\right)}-m_{0}\right)}^{⊤}V_{0}^{-1}\left(\theta_{1}^{\left(j\right)}-m_{0}\right)}{2}\right)\\ Y^{\left(j\right)} & |\sigma^{2(j)},\theta_{1}^{\left(j\right)}∼N_{n}\left(1_{n}\theta_{1}^{\left(j\right)},\sigma^{2(j)}I_{n}\right). \end{array}$$**Model (Equation 11) under A**$$\begin{array}{c} \sigma^{2(j)}∼IG(a_{0},b_{0})\\ \theta_{1}^{\left(j\right)}|\sigma^{2(j)}∼N\left(m_{0},\sigma^{2(j)}V_{0}\right)\\ Y^{\left(j\right)}|\sigma^{2(j)},\theta_{1}^{\left(j\right)}∼N_{n}\left(1_{n}\theta_{1}^{\left(j\right)},\sigma^{2(j)}I_{n}\right). \end{array}$$**Model (Equation 12) under H**$$\begin{array}{c} \theta_{2}^{\left(j\right)}=0\\ \theta_{1}^{\left(j\right)}|\theta_{2}^{\left(j\right)}=0∼t_{s}\left(2a_{0}+1;m_{0_{1.2}}\left(\theta_{2}^{\left(j\right)}\right),\frac{2b_{0}+{\left(\theta_{2}^{\left(j\right)}-m_{0_{2}}\right)}^{⊤}\;{V_{0}}_{22}^{-1}\left(\theta_{2}^{\left(j\right)}-m_{0_{2}}2\right)}{2a_{0}+1}V_{0_{11.2}}\right)\\ \sigma^{2(j)}|\theta_{1}^{\left(j\right)},\theta_{2}^{\left(j\right)}=0∼IG\left(a_{0}+1,b_{0}+\frac{{\left(\theta^{\left(j\right)}-m_{0}\right)}^{⊤}\;{V_{0}}^{-1}\left(\theta^{\left(j\right)}-m_{0}\right)}{2}\right)\\ Y^{\left(j\right)}|\sigma^{2(j)},\theta_{1}^{\left(j\right)},\theta_{2}^{\left(j\right)}=0∼N_{n}\left(X\theta^{\left(j\right)},\sigma^{2(j)}I_{n}\right). \end{array}$$**Model (Equation 12) under A**$$\begin{array}{cc} \sigma^{2(j)} & ∼IG(a_{0},b_{0})\\ \theta^{\left(j\right)} & |\sigma^{2(j)}∼N_{p}\left(m_{0},\sigma^{2(j)}V_{0}\right)\\ Y^{\left(j\right)} & |\sigma^{2(j)},\theta^{\left(j\right)}∼N_{n}\left(X\theta^{\left(j\right)},\sigma^{2(j)}I_{n}\right). \end{array}$$

Then, $y^{\left(j\right)}=\left(y_{1}^{\left(j\right)},\cdots y_{n}^{\left(j\right)}\right)$ is a random sample of the conditional distribution of Y, j=1⋯M.

In a first stage, we considered model (Equation 11) where $\theta =\theta_{1}$ and model (Equation 12) with $\theta ={\left(\theta_{1},\theta_{2}\right)}^{⊤}$. Note that the dimensionality of the parameter space, denoted by *d*, is different in the two models: for model (Equation 11), the dimensionality is d=2 and for model (Equation 12), the dimensionality is d=3. Samples of size M=1000 were generated for each model under the respective hypotheses and also for different sample sizes between n=10 and n=5000. In model (Equation 12), the covariate $x_{i1}$, i=1⋯n, was generated from a standard normal distribution. Finally, to obtain the adaptive values $\alpha_{\varphi^{*}}$ and $\beta_{\varphi^{*}}$, the two types of errors were considered as equally severe, that is, a=b=1.

Figure 1 shows the averaged error probabilities for the FBST as functions of *k* for a sample size n=100. This was replicated for all sample sizes in order to numerically find the corresponding $k^{*}$ value that minimizes $\alpha_{\varphi_{e}}+\beta_{\varphi_{e}}$. Table 1 and Table 2 and Figure 2 and Figure 3 present the $k^{*}$ and $\alpha_{\varphi_{P}^{*}}$ values as function of *n* for each model. As can be seen, both values have a decreasing trend when the sample size increases. In the case of the cutoff value for the evidence, it is possible to notice the differences in the results when the dimensionality of the parameter space change. Then, the $k^{*}$ value depends not only on the sample size but also on the dimensionality of the parameter space, more specifically, it is greater when *d* is higher. However, this does not occur with $\alpha_{\varphi_{P}^{*}}$, which maintains almost the same values even if *d* increases. On the other hand, Figure 4 and Figure 5 illustrate that in all these models, the optimal averaged error probabilities and their linear combination also decrease with increasing sample size.

We choose a single random sample $y_{0}$ to calculate the *e*-value and *P*-value for the models. Table 3 displays the results: the cases where H is rejected being represented by the cells in **boldface**. It can be observed that the decision remains the same regardless of the index used.

As the second stage in our simulation study, we set two sample sizes n=60 and n=120 to perform the tests for model (Equation 12), increasing the dimensionality of the parameter space. In that scenario, the vector of coefficients was such that $\theta ={\left(\theta_{1}^{⊤},\theta_{2}\right)}^{⊤}$ and the hypotheses to be tested were
$$\begin{array}{c} H:\theta_{2}=0\\ A:\theta_{2}\neq 0. \end{array}$$

So, by varying the dimension of vector $\theta_{1}$, the different models considered for each test were obtained. Table 4 and Table 5 and Figure 6 and Figure 7 show the $k^{*}$ and $\alpha_{\varphi_{P}^{*}}$ values as functions of *d*. For d=2, the values correspond to model (Equation 11). We can say that, for a fixed hypothesis, the larger the dimensionality of the parameter space, the greater the value of $k^{*}$. In the case of the $\alpha_{\varphi_{P}^{*}}$ value, it does not change significantly when the dimensionality of the parameter space increases, except when the number of parameters is very large in relation to the sample size.

Table 6 presents the *e*-value and *P*-value calculated for a single random sample $y_{0}$. Here, with the *e*-value the null hypothesis is less easily rejected. This may be related to two things: it may be due to approximation error as a result of the simulation process or due to the fact that the evidence apparently converges to 1 as the dimensionality of the parameter space increases, in which case a more detailed study is required.

## 5. Numerical Examples

In this section, we present two applications with real datasets. We choose $a_{0}=3$ and $b_{0}=2$ as parameters of the inverse gamma prior distribution for $\sigma^{2}$. Additionally, in the normal prior for θ given $\sigma^{2}$, $m_{0}=0_{p\times 1}$ and $V_{0}=I_{p}$ are taken as parameters. The Monte Carlo approximations were made generating samples of size *M* =10,000.

### 5.1. Budget Shares of British Households Dataset

We select a dataset that draws 1519 observations from the 1980–1982 British Family Expenditure Surveys (FES) ([25]). In our application, we want to fit the model
(13)$$y_{i}=\theta_{1}+\theta_{2}\;x_{i1}+\theta_{3}\;x_{i2}+\theta_{4}\;x_{i3}+\theta_{5}\;x_{i4}+\varepsilon_{i},\;\varepsilon_{i}∼N\left(0,\sigma^{2}\right).$$

We consider as explanatory variables, respectively, the total net household income (rounded to the nearest 10 UK pounds sterling) ($x_{1}$), the budget share for alcohol expenditure ($x_{2}$), the budget share for fuel expenditure, and the age of household head ($x_{3}$). We take the budget share for food expenditure as the dependent variable (y). All the expenditures and income are measured in pounds sterling per week.

Table 7 summarizes the results for the hypotheses $H:\theta_{j}=0$, j=1…5, by performing the test with the *p*-value at 0.05 significance level and also the *e*-value and the *P*-value with their respective adaptive significance levels. The cases where H is rejected are represented by the cells in **boldface**. ${\hat{\theta }}_{Freq}$ and ${\hat{\theta }}_{Bayes}$ are, respectively, the classical maximum likelihood estimator and the Bayes estimator of θ. It can be seen that unlike the *p*-value, the *e*-value and the *P*-value do not reject the hypothesis of nullity of the coefficient associated with the age of household head variable.

Table 8 exposes the optimal averaged error probabilities using the *e*-value and the *P*-value. It can be noted that the values are very similar with both methodologies. 

### 5.2. Boston Housing Dataset

We also take a dataset that contains information about housing values obtained from census tracts in the Boston Standard Metropolitan Statistical Area (SMSA) in 1970 ([26]). These data are composed of 506 samples and 14 variables. The regression model we use is
(14)$$y_{i}=\theta_{1}+\theta_{2}\;x_{i1}+\theta_{3}\;x_{i2}+\theta_{4}\;x_{i3}+\theta_{5}\;x_{i4}+\theta_{6}\;x_{i5}+\theta_{7}\;x_{i6}+\theta_{8}\;x_{i7}+\theta_{9}\;x_{i8}+\theta_{10}\;x_{i9}+\varepsilon_{i},$$
$$\varepsilon_{i}∼N\left(0,\sigma^{2}\right).$$

We choose the following explanatory variables to fit our model: per capita crime rate by town ($x_{1}$), the proportion of residential land zoned for lots over 25.000 sq. ft ($x_{2}$), the proportion of non-retail business acres per town ($x_{3}$), the proportion of non-retail business acres per town ($x_{4}$), the average number of rooms per dwelling ($x_{5}$), the proportion of owner-occupied units built prior to 1940 ($x_{6}$), the weighted mean of distances to five Boston employment centers ($x_{7}$), the full-value property tax rate per 10.000 ($x_{8}$), the pupil–teacher ratio by town, and $1000{\left(Bk-0.63\right)}^{2}$, where Bk is the proportion of black people by town ($x_{9}$). The dependent variable is the median value of the owner-occupied homes (in 1000 s) in the census tract (y).

The results for the hypotheses $H:\theta_{j}=0$, j=1…10 by performing the test with the *p*-value, the *e*-value and the *P*-value, are summarized in Table 9. In this case, with the *e*-value the null hypotheses are less rejected. The *e*-value does not reject the hypotheses of nullity of the coefficients associated with the proportion of residential land zoned for lots over 25.000 sq. ft and proportion of non-retail business acres per town variables, while the *p*-value does. On the other hand, the *P*-value, unlike the *p*-value, does not reject the hypothesis for the proportion of residential land zoned for lots over 25.000 sq. ft variable, but it does for the Intercept. As can be observed in Table 10, for these data, the optimal averaged error probabilities values are also very close.

## 6. Conclusions

In this work, we present a method to find a cutoff value $k^{*}$ for the Bayesian evidence in the FBST by minimizing the linear combination of the averaged type-I and type-II error probabilities for a given sample size *n* and also for a given dimensionality *d* of the parameter space in the context of linear regression models with unknown variance under the Bayesian perspective. In that sense, we provide a solution to the existing problem in the usual approach of hypothesis-testing procedures based on fixed cutoffs for measures of evidence: the increase of the sample size leads to the rejection of the null hypothesis. Furthermore, we compare our results with those obtained by using the test proposed by the authors of [13,14]. With our suggestion of cutoff value for the evidence in the FBST and also with the procedure proposed by the authors of [13,14], increasing the sample size implies that the probabilities of both kinds of optimal averaged errors and their linear combination decrease, unlike most cases, where, by setting a single level of significance independent of sample size, only type-II error probability decreases.

A detailed study is still needed for more complex models, so the methodology we propose to determine the adaptive cutoff value for evidence in the FBST could be extended to models with different prior specifications, which would involve, among other things, using approximate methods to find the prior predictive densities under the null and alternative hypotheses.   

## Figures and Tables

**Figure 1 entropy-25-00019-f001:**
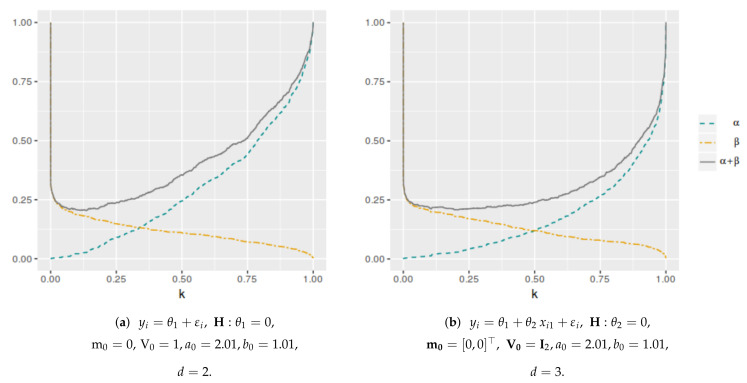
Averaged error probabilities ($\alpha_{\varphi_{e}}$, $\beta_{\varphi_{e}}$ and $\alpha_{\varphi_{e}}+\beta_{\varphi_{e}}$) as function of *k*. Sample size n=100.

**Figure 2 entropy-25-00019-f002:**
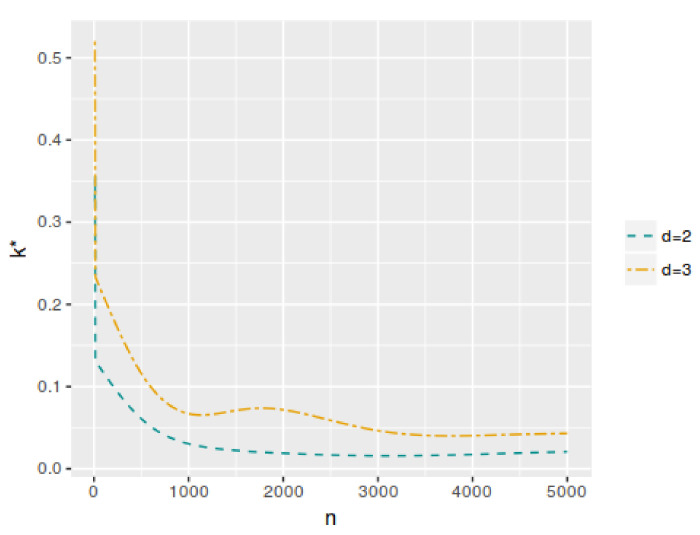
Cutoff values $k^{*}$ for $ev\left(H;y\right)$ as a function of *n*, with d=2 and d=3.

**Figure 3 entropy-25-00019-f003:**
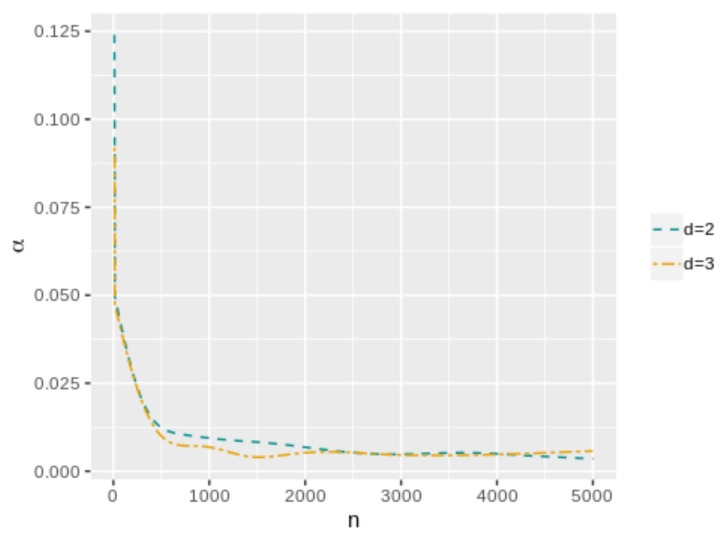
Optimal averaged type-I error probability ($\alpha_{\varphi^{*}}$) as a function of *n*, with d=2 and d=3.

**Figure 4 entropy-25-00019-f004:**
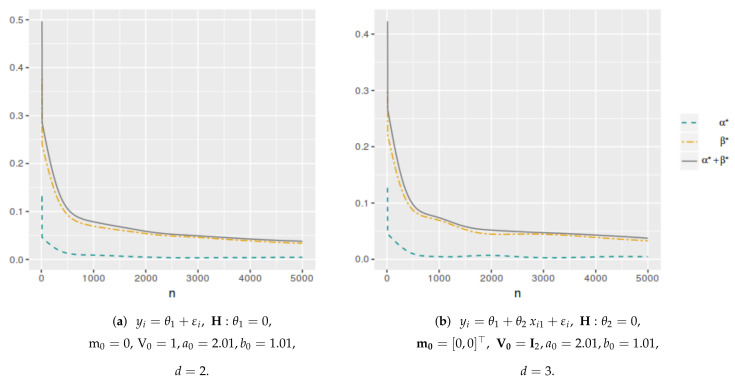
Unknown-variance model optimal averaged error probabilities ($\alpha_{\varphi_{e}^{*}}^{*}$, $\beta_{\varphi_{e}^{*}}^{*}$ and $\alpha_{\varphi_{e}^{*}}^{*}+\beta_{\varphi_{e}^{*}}^{*}$) as functions of *n*.

**Figure 5 entropy-25-00019-f005:**
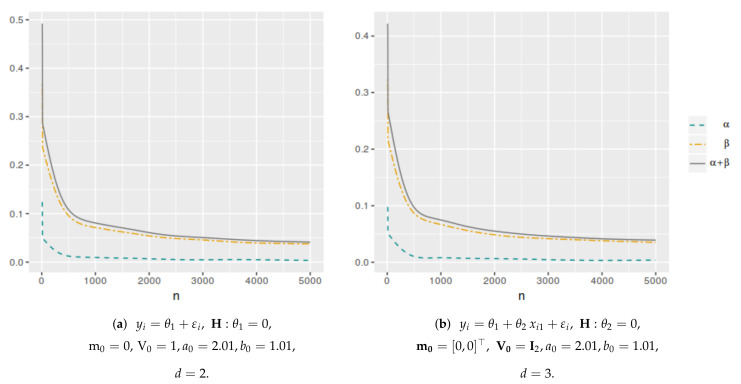
Optimal averaged error probabilities ($\alpha_{\varphi^{*}}$, $\beta_{\varphi^{*}}$ and $\alpha_{\varphi^{*}}+\beta_{\varphi^{*}}$) as functions of *n*.

**Figure 6 entropy-25-00019-f006:**
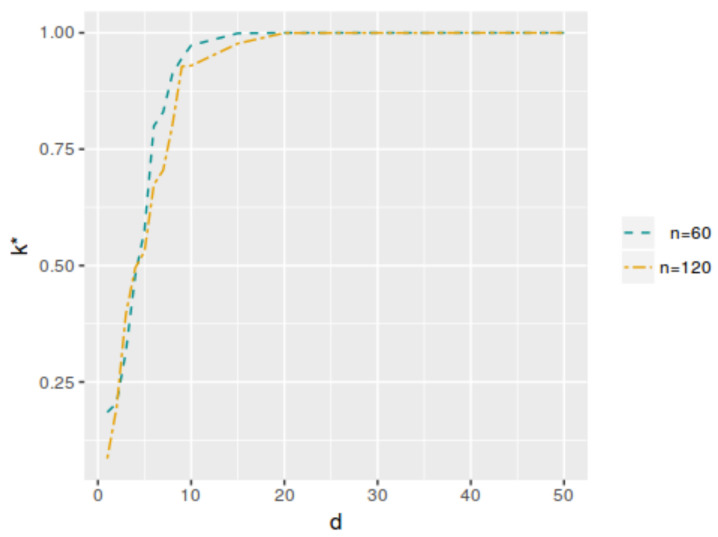
Unknown-variance model cutoff values $k^{*}$ for $ev\left(H;y\right)$ as a function of *d*, with n=60 and n=120.

**Figure 7 entropy-25-00019-f007:**
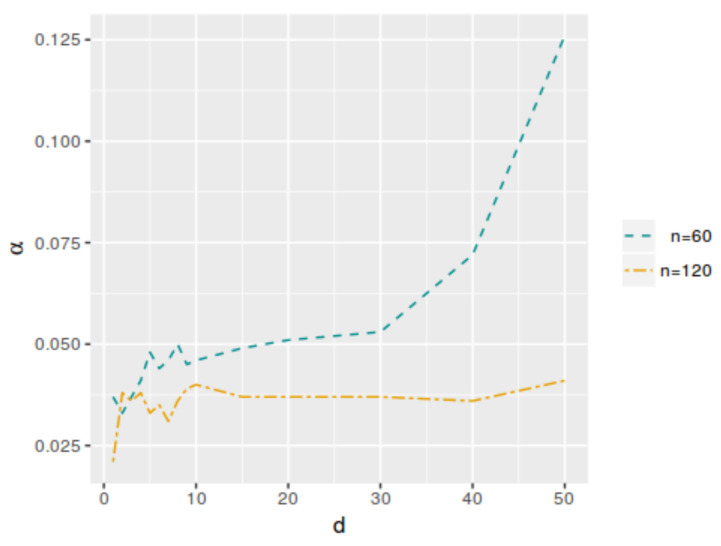
Optimal averaged type-I error probability ($\alpha_{\varphi^{*}}$) as a function of *d*, with n=60 and n=120.

**Table 1 entropy-25-00019-t001:** Cutoff values $k^{*}$ for $ev\left(H;y\right)$ as a function of *n*, with d=2 and d=3.

	$k^{*}$	
*n*	d=2	d=3
10	0.32530	0.51220
50	0.12534	0.22442
100	0.11705	0.21081
150	0.10889	0.19735
200	0.10092	0.18416
250	0.09323	0.17132
300	0.08587	0.15894
350	0.07893	0.14713
400	0.07243	0.13598
450	0.06641	0.12560
500	0.06091	0.11606
1000	0.03035	0.06689
1500	0.02223	0.07086
2000	0.01892	0.07173

**Table 2 entropy-25-00019-t002:** Optimal averaged type-I error probability ($\alpha_{\varphi^{*}}$) as a function of *n*, with d=2 and d=3.

	$\alpha_{\varphi^{*}}$	
*n*	d=2	d=3
10	0.12400	0.09200
50	0.04515	0.04327
100	0.03899	0.03775
150	0.03327	0.03252
200	0.02817	0.02772
250	0.02380	0.02341
300	0.02018	0.01963
350	0.01732	0.01642
400	0.01513	0.01376
450	0.01353	0.01163
500	0.01241	0.01002
1000	0.00941	0.00683
1500	0.00827	0.00398
2000	0.00681	0.00524

**Table 3 entropy-25-00019-t003:** Cutoff values $k^{*}$, $ev\left(H;y_{0}\right)$ and *P*-value (**y**_0_) as function of *n*, with d=2 and d=3.

d=2					d=3			
*n*	$k^{*}$	ev	$\alpha_{\varphi_{P}^{*}}$	Pv	$k^{*}$	ev	$\alpha_{\varphi_{P}^{*}}$	Pv
10	0.3253	0.9838	0.1240	0.7510	0.5122	0.9696	0.0920	0.4850
50	0.1253	**0.0820**	0.0451	**0.0190**	0.2244	0.9261	0.0433	0.3570
100	0.1171	**0.0000**	0.0390	**0.0000**	0.2108	0.4176	0.0377	0.0650
150	0.1089	**0.0973**	0.0333	**0.0200**	0.1974	0.2965	0.0325	0.0510
200	0.1009	**0.0036**	0.0282	**0.0000**	0.1842	**0.0466**	0.0277	**0.0040**
250	0.0932	**0.0001**	0.0238	**0.0000**	0.1713	**0.0620**	0.0234	**0.0050**
300	0.0859	**0.0000**	0.0202	**0.0000**	0.1589	**0.0119**	0.0196	**0.0010**
350	0.0789	**0.0000**	0.0173	**0.0000**	0.1471	**0.0282**	0.0164	**0.0030**
400	0.0724	**0.0000**	0.0151	**0.0000**	0.1360	**0.0347**	0.0138	**0.0020**
450	0.0664	**0.0000**	0.0135	**0.0000**	0.1256	**0.0628**	0.0116	**0.0040**
500	0.0609	**0.0000**	0.0124	**0.0000**	0.1161	**0.0181**	0.0100	**0.0010**
1000	0.0303	**0.0000**	0.0094	**0.0000**	0.0669	**0.0000**	0.0068	**0.0010**
1500	0.0222	**0.0000**	0.0083	**0.0000**	0.0709	**0.0000**	0.0040	**0.0010**
2000	0.0189	**0.0000**	0.0068	**0.0000**	0.0717	**0.0000**	0.0052	**0.0010**

**Table 4 entropy-25-00019-t004:** Unknown-variance model cutoff values $k^{*}$ for $ev\left(H;y\right)$ as a function of *d*, with n=60 and n=120.

	$k^{*}$	
*d*	n=60	n=120
2	0.18500	0.08560
3	0.20420	0.19480
4	0.31510	0.39630
5	0.47790	0.49500
6	0.57670	0.53040
7	0.79970	0.67400
8	0.82970	0.70490
9	0.91250	0.80310
10	0.94540	0.92770
11	0.97300	0.92940
21	0.99990	0.99960
31	0.99990	0.99970
41	0.99990	0.99990
51	0.99990	0.99990

**Table 5 entropy-25-00019-t005:** Optimal averaged type-I error probability ($\alpha_{\varphi^{*}}$) as a function of *d*, with n=60 and n=120.

	$\alpha_{\varphi^{*}}$	
*d*	n=60	n=120
2	0.03700	0.02100
3	0.03300	0.03800
4	0.03700	0.03600
5	0.04100	0.03800
6	0.04800	0.03300
7	0.04400	0.03500
8	0.04600	0.03100
9	0.05000	0.03600
10	0.04500	0.03900
11	0.04600	0.04000
21	0.05100	0.03700
31	0.05300	0.03700
41	0.07200	0.03600
51	0.12600	0.04100

**Table 6 entropy-25-00019-t006:** Cutoff values $k^{*}$, $ev\left(H;y_{0}\right)$ and *P*-value (**y**_0_) as functions of *d*, with n=60 and n=120.

n=60					n=120			
*d*	$k^{*}$	ev	$\alpha_{\varphi_{P}^{*}}$	Pv	$k^{*}$	ev	$\alpha_{\varphi_{P}^{*}}$	Pv
2	0.1850	0.6865	0.0370	0.3660	0.0856	**0.0082**	0.0210	0.0010
3	0.2042	0.5849	0.0330	0.1360	0.1948	0.7199	0.0380	0.1760
4	0.3151	0.8119	0.0370	0.1820	0.3963	0.9230	0.0360	0.2470
5	0.4779	**0.0000**	0.0410	**0.0000**	0.4950	**0.0000**	0.0380	**0.0010**
6	0.5767	**0.5672**	0.0480	**0.0290**	0.5304	0.7002	0.0330	0.0360
7	0.7997	0.8854	0.0440	0.0820	0.6740	0.9992	0.0350	0.2860
8	0.8297	**0.3267**	0.0460	**0.0050**	0.7049	0.7858	0.0310	**0.0260**
9	0.9125	**0.1919**	0.0500	**0.0020**	0.8031	**0.0009**	0.0360	**0.0010**
10	0.9454	**0.0006**	0.0450	**0.0010**	0.9277	**0.0001**	0.0390	**0.0010**
11	0.9730	**0.0000**	0.0460	**0.0000**	0.9294	**0.0000**	0.0400	**0.0000**
21	0.9999	**0.0000**	0.0510	**0.0000**	0.9996	**0.0000**	0.0370	**0.0000**
31	0.9999	1.0000	0.0530	**0.0240**	0.9997	**0.0495**	0.0370	**0.0010**
41	0.9999	**0.9998**	0.0720	**0.0010**	0.9999	**0.0004**	0.0360	**0.0010**
51	0.9999	1.0000	0.1260	**0.0000**	0.9999	**0.0000**	0.0410	**0.0000**

**Table 7 entropy-25-00019-t007:** Budget shares of British households dataset hypothesis-testing summary.

Coefficients	${\hat{\theta }}_{\mathrm{Freq}}$	α	pv	${\hat{\theta }}_{\mathrm{Bayes}}$	$k^{*}$	ev	$\alpha_{\varphi_{P}^{*}}$	Pv
Intercept	0.3758	0.0500	**0.0000**	0.3700	0.7078	**0.0000**	0.0382	**0.0000**
$x_{i1}$	−0.0004	0.0500	**0.0000**	−0.0004	0.0113	**0.0000**	0.0001	**0.0000**
$x_{i2}$	−0.1533	0.0500	**0.0003**	−0.1283	0.9410	**0.1890**	0.1278	**0.0172**
$x_{i3}$	0.1717	0.0500	**0.0007**	0.1487	0.9520	**0.1957**	0.1468	**0.0143**
$x_{i4}$	0.0009	0.0500	**0.0119**	0.0010	0.0764	0.3048	0.0004	0.0666

**Table 8 entropy-25-00019-t008:** Budget shares of British households dataset optimal averaged error probabilities.

Coefficients	$\alpha_{\varphi_{e}^{*}}^{*}$	$\alpha_{\varphi_{P}^{*}}$	$\beta_{\varphi_{e}^{*}}^{*}$	$\beta_{\varphi_{P}^{*}}$
Intercept	0.0466	0.0382	0.2157	0.2193
$x_{i1}$	0.0000	0.0001	0.0006	0.0006
$x_{i2}$	0.1521	0.1278	0.4146	0.4145
$x_{i3}$	0.1508	0.1468	0.4679	0.4410
$x_{i4}$	0.0004	0.0004	0.0080	0.0083

**Table 9 entropy-25-00019-t009:** Boston housing dataset hypothesis-testing summary.

Coefficients	${\hat{\theta }}_{\mathrm{Freq}}$	α	pv	${\hat{\theta }}_{\mathrm{Bayes}}$	$k^{*}$	ev	$\alpha_{\varphi_{P}^{*}}$	Pv
Intercept	1.7035	0.0500	0.6958	1.2035	0.9998	1.0000	0.1916	**0.0085**
$x_{i1}$	−0.1244	0.0500	**0.0006**	−0.1244	0.5780	**0.3365**	0.0010	**0.0001**
$x_{i2}$	0.0359	0.0500	**0.0224**	0.0362	0.4089	0.9012	0.0004	0.0025
$x_{i3}$	−0.1489	0.0500	**0.0235**	−0.1473	0.6390	0.9114	0.0025	**0.0023**
$x_{i4}$	6.7165	0.0500	**0.0000**	6.7336	0.9296	**0.0000**	0.0143	**0.0000**
$x_{i5}$	−0.0655	0.0500	**0.0000**	−0.0648	0.3275	**0.0141**	0.0001	**0.0000**
$x_{i6}$	−1.3198	0.0500	**0.0000**	−1.3091	0.8146	**0.0001**	0.0095	**0.0000**
$x_{i7}$	−0.0030	0.0500	0.2324	−0.0030	0.0124	0.9996	0.0002	0.0198
$x_{i8}$	−0.7652	0.0500	**0.0000**	−0.7528	0.8223	**0.0003**	0.0053	**0.0000**
$x_{i9}$	0.0145	0.0500	**0.0000**	0.0147	0.0297	**0.0113**	0.0001	**0.0000**

**Table 10 entropy-25-00019-t010:** Boston housing dataset optimal averaged error probabilities.

Coefficients	$\alpha_{\varphi_{e}^{*}}^{*}$	$\alpha_{\varphi_{P}^{*}}$	$\beta_{\varphi_{e}^{*}}^{*}$	$\beta_{\varphi_{P}^{*}}$
Intercept	0.1321	0.1916	0.6494	0.4946
$x_{i1}$	0.0018	0.0010	0.0165	0.0173
$x_{i2}$	0.0006	0.0004	0.0075	0.0079
$x_{i3}$	0.0030	0.0025	0.0286	0.0292
$x_{i4}$	0.0222	0.0143	0.1123	0.1181
$x_{i5}$	0.0000	0.0001	0.0068	0.0068
$x_{i6}$	0.0091	0.0095	0.0825	0.0808
$x_{i7}$	0.0000	0.0002	0.0016	0.0015
$x_{i8}$	0.0081	0.0053	0.0494	0.0521
$x_{i9}$	0.0000	0.0001	0.0019	0.0017

## Data Availability

The real datasets are freely available in the Ecdat package ([27]) (BudgetUK dataset) and the MASS package ([28]) (Boston dataset) of R software ([29]).

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
