# Peer review of "Adaptive Significance Levels in Tests for Linear Regression Models: The e-Value and P-Value Cases"

_entropy, 2022, doi:10.3390/e25010019_

Round 1

Reviewer 1 Report

See attached report.

Author Response

The reviewer forgot to upload the report and didn't answer to our email.

Reviewer 3 Report

This is an interesting and well written paper about FBST for linear regression models comparing e-value (under a Bayesian perspective) and p-values (frequentist one). Only a couple of minor comments:

1. Please indicate the expression of \hat\theta in page 4 line 106.

2. Along the text I don't found an interpretation of the values a and b in the linear combination of type-I and type-II errors. I assume that they represent the severity which both errors are considered. Please, the quantities a and b need to be clarified.

Round 2

Reviewer 1 Report

Overall the authors have done a nice job on the revsion and I'm happy to recommend acceptance. A couple of minor points, that don't require any modifications to the paper, are as follows.

1. I still believe that using a p-value based on Bayes factor to measure evidence is an approach that can result in contradictions as in there is evidence found against a hypothesis when the BF indicates evidence in favor. There is a good reason why 1 is used for the cut-off when using a BF to determine evidence for or against. As far as various scales used to determine the strength of the evidence provided by the BF, I've never seen a good universal argument for any of them and the context of the Jeffreys-Lindley paradox, where increasingly diffuse priors lead to the BF going to infinity, shows the arbitrariness of such scales. Basically BF's need to be calibrated and your p-value is probably one good way of doing this.

2. You are correct that I misspoke about the definition of the Bayes factor (my apologies).  I meant to say something along the following lines. The Bayes factor is the ratio of the posterior odds in favor to the prior odds in favor and that does result in the formula commonly used. But there are several reasons to question the validity of this expression in general as an appropriate measure of evidence. It *requires* that a discrete mass p>0 be placed on the null so in actuality the BF is not defined with a single continuous prior and a perfectly good prior needs to be modified. In such a context, if one wished to use the BF to compare the evidence for several different values, then effectively the prior needs to be changed for each value, which seems very inappropriate, and yet such comparisons seem quite natural as, for example, in estimation problems.

Just my thoughts on this.

Author Response

The authors thank the referee for his/her important comments on the manuscript that helped to improve it.

Reviewer 2 Report

It appears that I omitted to upload my original review. Apologies for that. I have (I hope) uploaded that review here -- as you will see, there are fundamental flaws in the methods and purpose of this manuscript.

Round 3

Reviewer 2 Report

The authors have made some very reasonable comments regarding this work, and my review. However, the criticisms made in my review remain (unfortunately) correct, and the authors response and amendments in no way fix the basic problems with this manuscript. I repeat that this is is not a significant contemporary contribution, not a Bayesian analysis, and not a suitable fit for Entropy.

To address a few of the authors' points: I concur that the Lebesgue measure is typically used by default, and without mention. But the authors are incorrect to say that this is always the case. Two significant examples I can think of are: function-space problems where the Lebesgue measure is not normalizable hence not possible as a reference measure, and the Jeffereys' priors (that I alluded to in my review), that are now referred to as reference priors or objective measures. The authors method depends on the value of the density function, hence is not a Bayesian method. In particular, it does not depend solely on the distribution, as the authors (incorrectly) claim.  Rather this manuscript reports a Bayes-inspired procedure, with applicability and results very dependent on the application.

The authors mentioned significant interest in a statistics journal on the topic considered. I repeat my suggestion that this manuscript be submitted to a statistics journal so the merit in statistics can be evaluated, though I suspect this manuscript would not get a good reception.